# Tuning Electrochemical Hydrogen-Evolution Activity of CoMoO$_4$ through Zn Incorporation

**Sanaz Chamani** [1,†], **Ebrahim Sadeghi** [1,2,†], **Ugur Unal** [3,4], **Naeimeh Sadat Peighambardoust** [1]
**and Umut Aydemir** [1,4,*]

1    Koç University Boron and Advanced Materials Applications and Research Center (KUBAM), Sariyer, Istanbul 34450, Turkey; schamani@ku.edu.tr (S.C.); esadeghi19@ku.edu.tr (E.S.); npeighambardoust@ku.edu.tr (N.S.P.)

2    Graduate School of Sciences and Engineering, Koç University, Sariyer, Istanbul 34450, Turkey

3    Koç University Surface Science and Technology Center (KUYTAM), Sariyer, Istanbul 34450, Turkey; ugunal@ku.edu.tr

4    Department of Chemistry, Koç University, Sariyer, Istanbul 34450, Turkey

\*    Correspondence: uaydemir@ku.edu.tr

†    These authors contributed equally to this work.

**Abstract:** Designing cheap, efficient, and durable electrocatalysts on three-dimensional (3D) substrates such as nickel foam (NF) for the hydrogen-evolution reaction (HER) is in high demand for the practical application of electrochemical water splitting. In this work, we adopted a simple one-step hydrothermal method to realize the incorporation of Zn into the lattice of CoMoO$_4$ with various atomic concentrations—Co$_{1-x}$Zn$_x$MoO$_4$ ($x$ = 0, 0.1, 0.3, 0.5, and 0.7). The morphological studies demonstrated that parent CoMoO$_4$ consists of nanoflowers and nanorods. However, as the concentration of Zn increases within the host CoMoO$_4$, the portion of nanoflowers decreases and simultaneously the portion of nanorods increases. Moreover, the substitution of Zn$^{2+}$ in place of Co$^{2+}$/Co$^{3+}$ creates oxygen vacancies in the host structure, especially in the case of Co$_{0.5}$Zn$_{0.5}$MoO$_4$, giving rise to lower charge-transfer resistance and a higher electrochemically active surface area. Therefore, among the prepared samples, Co$_{0.5}$Zn$_{0.5}$MoO$_4$ on NF showed an improved HER performance, reaching 10 mA cm$^{-2}$ at an overpotential as low as 204 mV in a 1.0 M KOH medium. Finally, the Co$_{0.5}$Zn$_{0.5}$MoO$_4$ electrode exhibited robust long-term stability at an applied current density of 10 mA cm$^{-2}$ for 20 h. The Faradaic efficiency determined by a gas chromatograph found that the hydrogen-production efficiency varied from 94% to 84%.

**Keywords:** electrocatalysis; hydrogen-evolution reaction; transition metal oxides; metal substitution

## 1. Introduction

The universally rapid consumption of traditional fossil fuels and the pernicious environmental impacts they create have encouraged researchers to actively and urgently seek a sustainable, renewable, and environmentally friendly energy source as an alternative [1,2]. Electrocatalytic water splitting is a potential approach to convert renewable electricity—generated from renewable energy resources—to hydrogen as a pure, viable, and carbon dioxide-free energy source [3,4]. The efficiency of the water-splitting process is closely dependent on the degree of competence of the electrocatalysts to drive the evolution of the hydrogen (HER) and oxygen (OER) from the medium [5,6]. Noble metal-based electrocatalysts, despite their robust activity, are not sufficiently durable. Additionally, these materials are expensive and rare [7]. Therefore, to circumvent these issues, great efforts have been made to design and develop catalysts based on earth-abundant transition metals (TMs) (e.g., Ni, Co, Fe, Zn, Cu, Mo) [8]. A lot of recent publications have proved the excellent performance of TM-based compounds in water electrolysis. Among these

compounds, transition metal oxides (TMOs), because of their low cost, rich redox properties, abundance, and durability under alkaline conditions, are one of the most promising candidates to consider [9,10].

Recently, there has been a surge of interest in the electrochemical properties of bimetallic oxides in comparison to single-component oxides due to their improved characteristics [11–13]. Bimetallic oxide catalysts could alter the electronic structures of the catalysts by regulating the composition, physical, and chemical aspects of the metals in the structure [14,15]. Moreover, bimetallic oxides offer multiple active sites, trap inactive active sites, and elevate the conductivity at the same time [14,16]. TM molybdates, as a subgroup of bimetallic compounds, have received much attention because of their high electrochemical activities, which can be ascribed to their rich polymorphism and high conductivity originating from metal molybdates [12,17]. $CoMoO_4$ is considered a potentially good electrocatalytic material, owing to the synergy between the redox ability of Co and the high adsorption capacity of Mo in its framework [12,18–20]. Yet, $CoMoO_4$ shows limited HER performance in alkaline media, which is most probably attributed to its poor intrinsic conductivity. Thus, it is a significant task to modify the electrocatalytic features of $CoMoO_4$ to act effectively in the alkaline HER [21].

Multiple strategies have been introduced to promote the conductivity and electrocatalytic properties of materials, including phase transition [22,23], coupling with highly conductive carbon materials, doping with heteroatoms [24], strain engineering [21,25], facet control [26], and oxygen vacancies [27,28]. Among them, elemental doping is a common way to optimize the physicochemical properties of catalysts and thereby adjust the electronic structure, which could availably improve HER performance [29–31]. Only a few reports have been released that elaborate on the effect of elemental doping on the HER activity of $CoMoO_4$, and those which have done so mostly focused on anion doping. Wen et al. [14] utilized a phosphorous doping technique to fabricate a controlled catalyst P-$CoMoO_4$@Ni with a hierarchical structure with different morphologies on the surface of nickel foam (NF). The best HER performances reported by this group were obtaining current densities of 100 and 150 mA cm$^{-2}$ under low overpotentials of 98 and 162 mV, respectively. Jiao et al. [32] took advantage of phosphorous doping using a facile hydrothermal method followed by low-temperature phosphidation to modulate the electronic structure of $CoMoO_4$ on NF for an alkaline HER. The developed P-$CoMoO_4$/NF demonstrated electrocatalytic activity toward the HER in 1.0 M KOH with a low overpotential of 89 mV at 10 mA cm$^{-2}$.

In this work, we report the effect of Zn substitution in the Co sites of monoclinic $CoMoO_4$ on the HER in a 1.0 M KOH medium. Ling et al. [24] conducted a thorough theoretical and experimental investigation on the electronic structure modulation of CoO nanorods by dual Ni and Zn doping for the HER. Based on combined density functional theory (DFT) calculations and careful microscopic and spectroscopic characterizations, they showed that the Zn dopants distribute inside the host oxide and control bulk electronic structure to promote electrical conduction. In addition, introducing a Zn dopant with a slightly larger atomic radius compared to Co into the host, $CoMoO_4$, and different electronic configurations will tailor the structural properties along with the chemical environment. Furthermore, comparing the nearly identical ionic radii of $Co^{2+}$ (0.063 nm) and $Zn^{2+}$ (0.074 nm) as well as their similar electronegativities—1.88 for Co and 1.65 for Zn—indicates the suitability of the Zn dopant in the host $CoMoO_4$ [33]. The Zn dopant with varying atomic concentrations was substituted using a simple one-pot hydrothermal technique in the final electrode, $Co_{1-x}Zn_xMoO_4$, where $x$ = 0.1, 0.3, 0.5, and 0.7. The effect of varied Zn substitution for the alkaline HER is discussed. Among these catalysts, $Co_{0.5}Zn_{0.5}MoO_4$ displayed the highest HER performance, with an overpotential of 204 mV at 10 mA cm$^{-2}$ and a Tafel slope of 162 mV dec$^{-1}$ in 1.0 M KOH.

## 2. Results and Discussion

### 2.1. Chemical Characterization

The schematic growth of $Co_{1-x}Zn_xMoO_4$ ($x$ = 0.0, 0.1, 0.3, 0.5, 0.7) on NF can be visualized in Figure 1a (further details can be found in Materials and Methods, Section 3.2). The XRD was performed on the as-prepared materials. The XRD patterns of $Co_{1-x}Zn_x MoO_4$ and the theoretical pattern of $CoMoO_4$ are illustrated in Figure 1b. The characteristic XRD peaks at 2θ of 13.10, 18.95, 23.12, 26.40, 28.34, 31.96, 33.47, 36.62, 38.66, 40.10, 43.45, 45.1, and 47.07 correspond to the respective crystalline planes of the (001), ($\bar{2}$01), (021), (220), ($\bar{3}$11), (112), ($\bar{2}$22), (400), (040), (330), ($\bar{4}$22), ($\bar{5}$11), and (421) and are well-matched with the monoclinic structure of β-$CoMoO_4$ ((ICDD No. 04-017-6377). It is pertinent to mention that the dashed line around 2θ of 26.40° indicates a noticeable peak shift toward smaller 2θ values with respect to the increase in the concentration of Zn in the host $CoMoO_4$, confirming the successful substitution of Zn and thereby implying an enlargement in lattice parameters (a better view of the XRD shift for (220) can be found in Figure S1, in the Supplementary Materials). Figure 1c depicts the crystal structure design of β-$CoMoO_4$, which crystallizes in the space group $C2/m$ (No.12). In this scheme, $CoO_6$ octahedrons are in purple and the tetrahedrons $MoO_4$ are in a gray color [34].

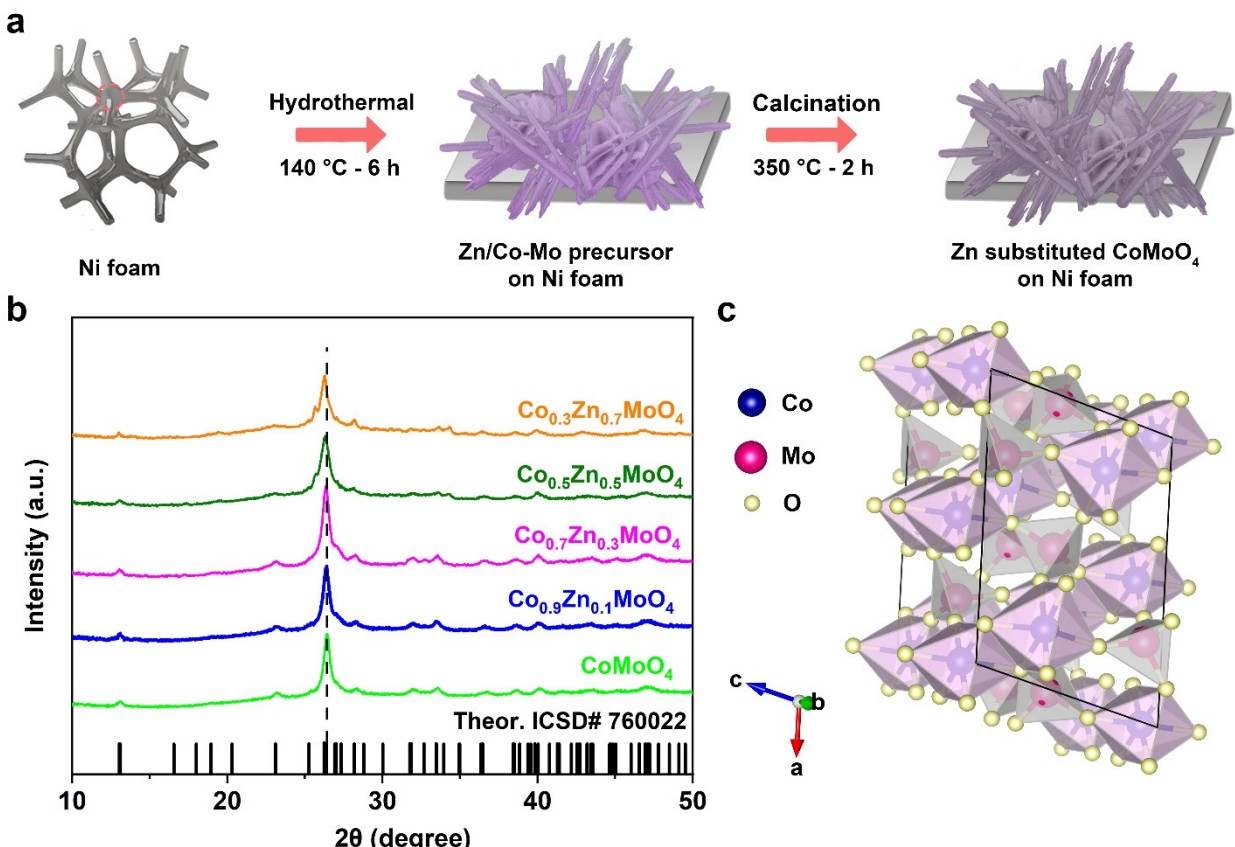

**Figure 1.** (**a**) Schematic illustration of preparation procedure of $Co_{1-x}Zn_xMoO_4$, (**b**) XRD patterns of the prepared $Co_{1-x}Zn_xMoO_4$ samples, and (**c**) Crystal structure of $CoMoO_4$.

An FE-SEM equipped with EDS, TEM, and HR-TEM was employed to study the surface morphology, analyze the elemental composition, and identify the interplanar spacing of as-prepared materials. Figure 2a–e displays the corresponding SEM images of pure $CoMoO_4$ and Zn-substituted $CoMoO_4$. The morphology of the catalysts grown on the NF substrate is a combination of two structures of nanoflowers and nanorods. The SEM microstructure of $CoMoO_4$ in Figure 2a demonstrates that the portion of nanoflowers and nanorods on the NF are somehow equal. However, the comparison of the SEM

image of $Co_{0.9}Zn_{0.1}MoO_4$ in Figure 2b with that of $CoMoO_4$ indicates that the size of nanoflowers reduced. Moreover, as the concentration of Zn increases within the host $CoMoO_4$, the portion of nanoflowers decreases and simultaneously the portion of nanorods increases. With this in mind, materials containing a higher concentration of Zn mostly consist of nanorods (refer to high-magnification SEM photographs shown in Figure 2d,e). The diameter of the nanorods was estimated to be between 100 and 300 nm. However, the length of the nanorods can reach a couple of hundreds of micrometers. The uniform growth of all as-obtained catalysts on the NF can be seen in Figure S2. Additionally, the main ingredients of the prepared materials (Mo, Co, O, and Zn) were verified through EDS analysis (the details can be found in Figure S3 and Table S1).

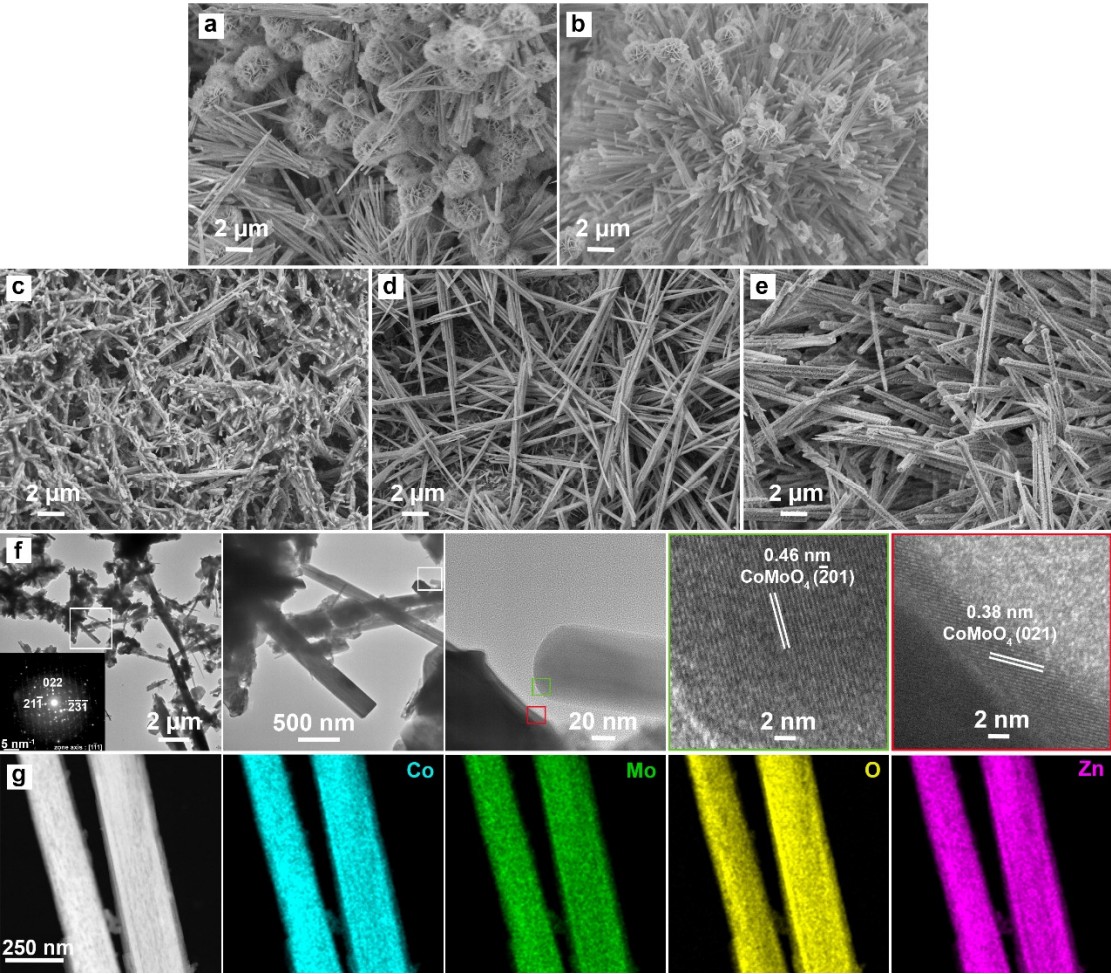

**Figure 2.** Morphological, structural, and microstructural characterizations of the samples. SEM images of (**a**) $CoMoO_4$, (**b**) $Co_{0.9}Zn_{0.1}MoO_4$, (**c**) $Co_{0.7}Zn_{0.3}MoO_4$, (**d**) $Co_{0.5}Zn_{0.5}MoO_4$, and (**e**) $Co_{0.3}Zn_{0.7}MoO_4$, respectively, (**f**) TEM photographs related to $Co_{0.5}Zn_{0.5}MoO_4$ (inset: SAED pattern), white square showing the high-magnification TEM images of nanorods, green and red squares showing the HR-TEM images of lattice spacing of $Co_{0.5}Zn_{0.5}MoO_4$, and (**g**) HAADF-STEM image and the corresponding EDS–STEM mappings of $Co_{0.5}Zn_{0.5}MoO_4$.

The observation of the TEM micrographs indicated that for the $Co_{0.5}Zn_{0.5}MoO_4$ sample, the dominating morphology is the nanorod-like structure (Figure 2f). The HR-TEM image of $Co_{0.5}Zn_{0.5}MoO_4$ revealed that the interplanar spacing of the $Co_{0.5}Zn_{0.5}MoO_4$ nanorods is about 0.46 nm (Figure 2f, fourth image from left) and 0.384 nm (Figure 2f, fifth image from left), which correspond to the ($\bar{2}01$) and (021) planes of monoclinic $CoMoO_4$, respectively. Furthermore, the HAADF-STEM image taken from the selected sample confirmed the even distribution of Co, Zn, Mo, and O on the random nanorods (see Figure 2g).

Fourier transform infrared (FT-IR) characterization was utilized to further verify the successful incorporation of Zn into Co sites in $CoMoO_4$. Figure S4 displays the FT-IR spectra of all samples. The absorption signals that emerged at 928 and 783 cm$^{-1}$ denote the stretching vibration mode of the O–Mo–O bond [35–38]. The peak located at 696 cm$^{-1}$ was attributed to the vibration of the Co–O–Mo group [37]. The band that appeared at 408 cm$^{-1}$ represents the Mo–O and $CoO_6$ groups of $CoMoO_4$ [39,40]. The absorption bands in the region of 500–1000 cm$^{-1}$ demonstrate shifts in the wavenumber as the content of Zn changes in the sample. This shift toward higher wavenumber values—blueshift—is an indication of the substitution of $Zn^{2+}$ for $Co^{2+}$, which increases monotonically with the increase of Zn concentration [41,42]. As a result, the incorporation of Zn atoms into the $CoMoO_4$ lattice strengthens the metal–oxygen bonds in the structure.

The surface chemical composition and oxidation states of prepared catalysts were characterized by XPS. The presence of expected elements was verified by the XPS survey shown in Figure S5a. The high-resolution XPS spectra of $CoMoO_4$ and $Co_{0.5}Zn_{0.5}MoO_4$ are depicted in Figure 3. From Figure 3a, we note how the Co 2p spectrum in $CoMoO_4$ revealed two major doublets and their corresponding shake-up satellite peaks. The lower BEs centered at 780.68 and 784.59 eV can be related to Co $2p_{3/2}$, while those located at higher BEs (796.71 and 800.20 eV) are correlated with Co $2p_{1/2}$. The Co 2p peak analysis shows that Co exists in oxidation states of 2+ and 3+ [9]. The high-resolution Mo 3d spectrum can be divided into two sets of doublets, which are ascribed to Mo $3d_{3/2}$ at higher BEs (235.03 and 236.79 eV) and Mo $3d_{5/2}$ at lower BEs (231.81 and 233.77 eV). These peaks confirm the presence of both oxidation states of $Mo^{6+}$ and $Mo^{4+}$ [43] on the surface. The O 1s XPS spectrum for $CoMoO_4$ also displayed three peaks at BEs—530.14, 533.54, and 537.06 eV—which can be attributed to the metal–oxygen, oxygen vacancy, and adsorbed hydroxyl groups on the surface [44,45].

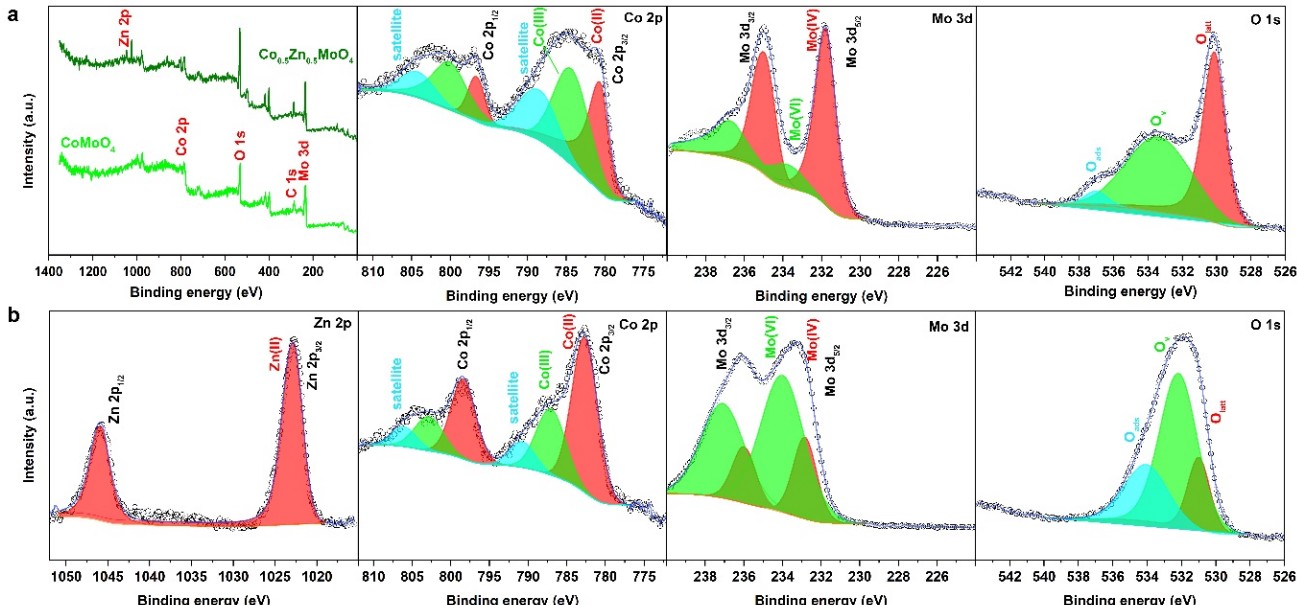

**Figure 3.** Spectroscopic characterizations of the samples (**a**) From left to right XPS survey of $CoMoO_4$ and $Co_{0.5}Zn_{0.5}MoO_4$, XPS Co 2p, Mo 3d, and O 1s for $CoMoO_4$, (**b**) From left to right, XPS Zn 2p, Co 2p, Mo 3d, and O 1s for $Co_{0.5}Zn_{0.5}MoO_4$.

The deconvolution of relevant XPS spectra for $Co_{0.5}Zn_{0.5}MoO_4$ (shown in Figure 3b) exhibited that Zn 2p species are composed of two spin-orbit doublets of Zn $2p_{3/2}$ and Zn $2p_{1/2}$, suggesting the substitution of $Zn^{2+}$ into the host structure (refer to Table S2). In addition, an evident peak shift was observed for both Co 2p and Mo 3d toward higher BEs compared with $CoMoO_4$. The substitution of $Zn^{2+}$ in place of $Co^{2+}/Co^{3+}$ gives rise to a mitigation of positive charge, resulting in the formation of more oxygen vacancies to

maintain the charge neutrality in the structure of $Co_{0.5}Zn_{0.5}MoO_4$ and therefore higher oxidation states on the metal sites [46]. It can be concluded that the Zn-site stabilizes the Co-site by oxidizing the Co to receive an improved structural sturdiness without experiencing significant deformation [46,47]. Similar to $CoMoO_4$, the core-level O 1s XPS spectrum for $Co_{0.5}Zn_{0.5}MoO_4$ was analyzed and three peaks were obtained at 530.07, 532.17, and 534.12 eV, corresponding to the identical oxygen-bearing groups. As expected, the amount of oxygen vacancy increased in $Co_{0.5}Zn_{0.5}MoO_4$ relative to unsubstituted $CoMoO_4$. The XPS spectra of all elements for each sample can be observed in Figure S5b–e. The subtle dissimilarity between the O 1s spectra of $Co_{0.5}Zn_{0.5}MoO_4$ and other electrodes stems from the larger oxygen vacancy peak for $Co_{0.5}Zn_{0.5}MoO_4$. It is worth noting that a correlation exists between the concentration of Zn substitution and the degree of XPS shift toward higher BEs for the Co 2p, Mo 3d, and Zn 2p spectra in all samples except for $Co_{0.3}Zn_{0.7}MoO_4$. As the concentration of Zn increases from $x = 0.1$ to $x = 0.5$, the XPS peaks shift to higher BEs compared to $CoMoO_4$ without substitution. Among the samples, $Co_{0.9}Zn_{0.1}MoO_4$ and $Co_{0.5}Zn_{0.5}MoO_4$ exhibited the least and most peak shift in the direction of higher BEs, respectively. Nonetheless, the trend is disrupted by the increase of Zn concentration to $x = 0.7$. It can be calculated that until $x = 0.5$, the structure possesses the highest amount of oxygen vacancy with high structural integrity. The creation of oxygen vacancies can modulate the electronic structure, boost conductivity, and enhance catalytic performance [48].

### 2.2. Electrocatalytic Hydrogen Evolution

The electrochemical properties of as-prepared catalysts for the HER were examined on NF with a ~1 mg cm$^{-2}$ catalyst loading as a working electrode in 1.0 M KOH solution.

Moreover, Figure 4a displays the electrocatalytic HER activities of as-prepared materials on the NF substrate along with blank NF and commercial Pt/C (20 wt%) as a comparison. Based on the results, the overpotentials (measured at 10 mA cm$^{-2}$) of the materials exhibit the following trend: $Pt/C < Co_{0.5}Zn_{0.5}MoO_4 < Co_{0.3}Zn_{0.7}MoO_4 < Co_{0.7}Zn_{0.3}MoO_4 < Co_{0.9}Zn_{0.1}MoO_4 < CoMoO_4 <$ blank NF (Figure 4b). A quick examination of this trend shows that the state-of-the-art Pt/C@NF electrode and blank NF, respectively, featured the lowest and highest overpotential at 10 mA cm$^{-2}$. Among the as-prepared materials, $CoMoO_4$ has the lowest performance and the highest overpotential. On the other hand, $Co_{0.5}Zn_{0.5}MoO_4$ showcased the best performance and the lowest overpotential. From the results it can be concluded that Zn incorporation modified the HER electrocatalysis, explaining that all Zn-substituted materials have better activity compared to their unsubstituted counterpart. To compare the results obtained in this work, the HER performances of previous studies have been tabulated in Table S3 [18,20,49–55].

To explore the electrochemical kinetic behavior, the Tafel slopes were calculated. The Tafel slopes of as-obtained materials are depicted in Figure 4c. The HER kinetics, however, are facile irrespective of the specified overpotential of the catalyst. In this sense, it can be inferred that all reported catalysts have quite identical HER kinetics [9]. In addition, the kinetics of the HER in alkaline media can be realized by the mechanisms of adsorbing and desorbing hydrogen atoms or molecular hydrogen via either the Volmer–Heyrovsky or Volmer–Tafel pathways [56]. The underlying reactions are:

$$H_2O + e^- + * \rightarrow H_{ads}* + OH^- \text{ (Volmer step)}$$

$$H_{ads}* + H_2O + e^- \rightarrow H_2 + OH^- + * \text{ (Heyrovsky step)}$$

$$H_{ads}* + H_{ads}* \rightarrow H_2 + 2* \text{ (Tafel step)}$$

where * denotes the available active sites and $H_{ads}*$ indicates the presence of atomic hydrogen at the active site. To understand the mechanism of the HER, Tafel plots obtained from LSV curves can be used as a means of inference.

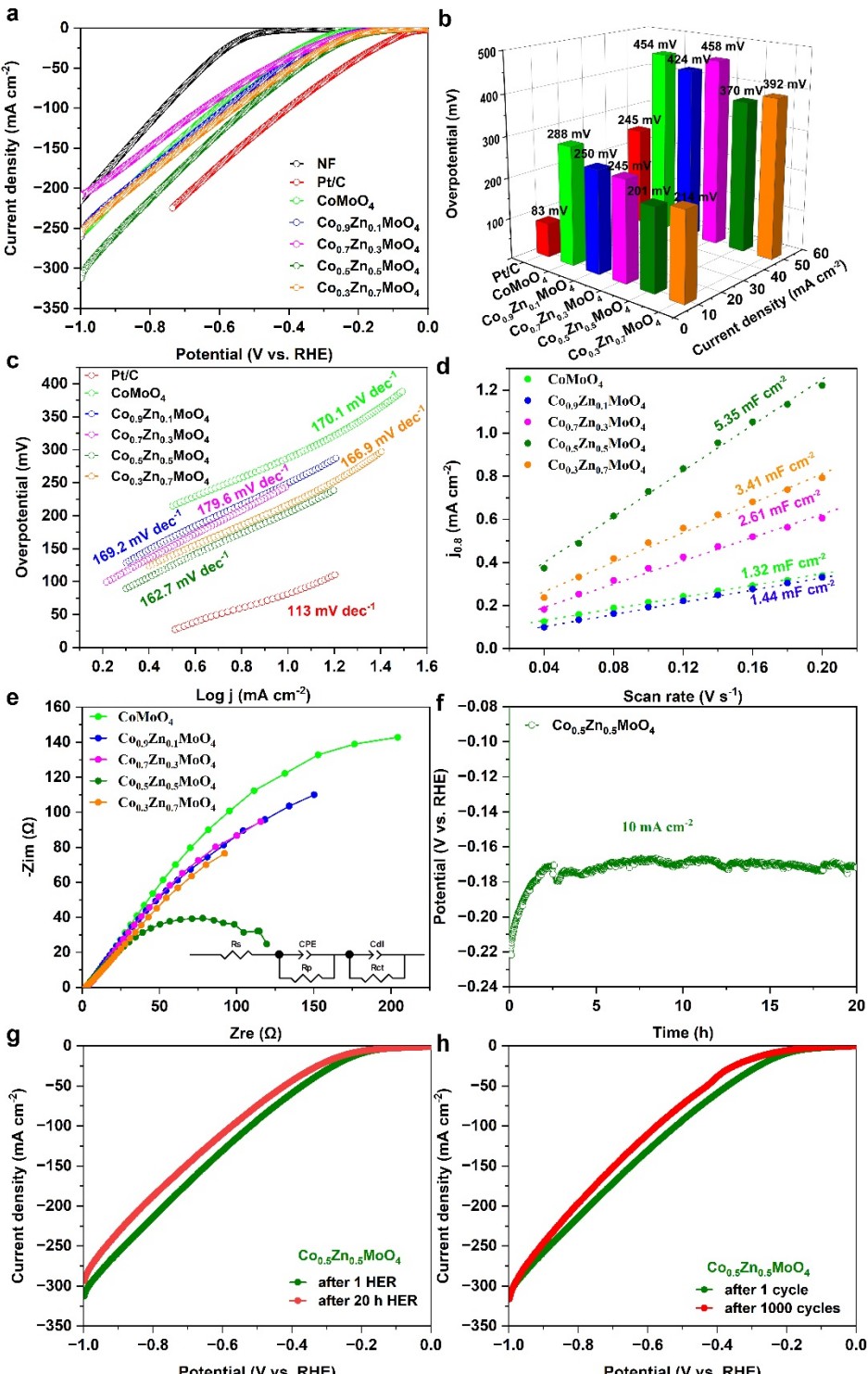

**Figure 4.** Electrocatalytic hydrogen-evolution performances. (**a**) HER polarization curves of the samples, (**b**) 3D bar graphs of overpotentials at 10 and 50 mA cm$^{-2}$, (**c**) HER Tafel plots obtained by polarization curves, (**d**) Capacitive current density versus scan rate curves for ECSA measurements, (**e**) EIS spectra recorded at 0 V versus RHE, (**f**) Long-term HER stability test of Co$_{0.5}$Zn$_{0.5}$MoO$_4$ at an applied current density of 10 mA cm$^{-2}$, (**g**) HER polarization curves of best-performing Co$_{0.5}$Zn$_{0.5}$MoO$_4$ before and after long-term HER, and (**h**) HER polarization curves of Co$_{0.5}$Zn$_{0.5}$MoO$_4$ before and after 1000 CV scans at 100 mV s$^{-1}$.

The reported Tafel slopes for the Volmer, Heyrovsky, and Tafel steps are 120, 40, and 30 mV dec$^{-1}$, respectively. Among the prepared samples, the commercial Pt/C catalyst shows the smallest Tafel slope value (113 mV dec$^{-1}$), suggesting that the Volmer reaction governs the overall HER process [57]. According to Figure 4c, all as-prepared materials showed Tafel slope values beyond 120 mV dec$^{-1}$, implying that the rate of the reaction is determined by the adsorption of hydrogen atoms on the active sites. The $Co_{0.5}Zn_{0.5}MoO_4$ sample showed a lower Tafel slope (162.7 mV dec$^{-1}$), signifying faster kinetics for hydrogen adsorption at active sites compared to other samples. In order to examine the reason for the high activity of $Co_{0.5}Zn_{0.5}MoO_4$, the ECSA was estimated with respect to $C_{dl}$ for these materials [55]. As there is a direct relation between ECSA and $C_{dl}$, the higher the $C_{dl}$ value, the better the electrocatalytic activity. The $Co_{0.5}Zn_{0.5}MoO_4$ gave the maximum $C_{dl}$ value (5.35 mF cm$^{-2}$), which is consistent with the HER activity (refer to Figure 4d). Therefore, the larger electroactive surface area is a favorable parameter for enhancing the electron transfer rate. The recorded CV scans within the potential window of 0.75–0.85 V vs. RHE for the as-obtained catalysts can be visualized in Figure S6.

To supplement the data from the electroactive surface area, we analyzed the BET surface area. The results showed that even though the BET values are relatively comparable, $Co_{0.5}Zn_{0.5}MoO_4$ possesses the highest BET surface area (10.95 m$^2$/g). The higher surface area offers more available active sites for the adsorption of hydrogen atoms, encourages fast charge transfer, and in turn, boosts electrocatalysis [58]. The typical $N_2$ adsorption–desorption isotherms are presented in Figure S7. Further details derived from BET analysis are tabulated in Table S4. To further unmask the HER kinetics, EIS measurements were implemented. Figure 4e presents the Nyquist plots of all fabricated samples on the NF substrate. The smallest and largest semicircles belong to $Co_{0.5}Zn_{0.5}MoO_4$ and $CoMoO_4$, which describe the degree of charge-transfer resistance ($R_{Ct}$) between electrolyte and electrode. This means that charge transfer favored the $Co_{0.5}Zn_{0.5}MoO_4$, leading to a facilitated kinetic barrier, and in turn improved electrocatalytic performance [59]. It must be pointed out that the inset image of Figure 4e symbolizes the equivalent circuit model for the charge-transfer path on the catalysts. In this model, $R_s$ denotes resistance induced by the solution and $R_p$ indicates resistance coming from the porous substrate. In addition, to fulfill the non-ideal behavior of the capacitive elements, a constant phase element (CPE) was applied. Similarly, the double-layer capacitance in the interface of electrode/solution was taken into account through $C_{dl}$. Further details associated with the charge transfer on the catalysts can be found in Table S5.

From a perspective of practical application, the HER long-term durability of $Co_{0.5}Zn_{0.5}MoO_4$ was evaluated by performing chronopotentiometry measurement at the constant current density of 10 mA cm$^{-2}$ in 1.0 M KOH. As presented in Figure 4f, no observable degradation was discerned after 20 h of continuous HER, demonstrating the robust tolerance of $Co_{0.5}Zn_{0.5}MoO_4$ in an alkaline solution. It is worth noting that the potential decreases within the first 2 h, which might be on account of changes in the oxidation states of transition metals and the exfoliation of nanostructures on the surface, giving rise to abundant active sites on the electrode. The LSV curve after 20 h disclosed that the HER overpotential at 10 mA cm$^{-2}$ increased from 204 mV to 248 mV (Figure 4g). Moreover, the long-term stability of $Co_{0.5}Zn_{0.5}MoO_4$ was assessed by 1000 CV scans over the reduction potential window. The HER after 1000 CV cycles was obtained as 236 mV at 10 mA cm$^{-2}$, confirming its favorable endurance (Figure 4h).

### 2.3. Post Electrolysis

To spot any changes in the morphology and chemical composition on the surface after long-term stability, the $Co_{0.5}Zn_{0.5}MoO_4$ electrode was checked by FE-SEM, TEM, and XPS. Figure 5a shows the post-electrocatalysis FESEM image of the catalyst on the NF. From the high-magnification SEM photograph, we can see that the nanorod morphology of the catalyst remained largely unchanged. However, the smooth nanorods transformed into a rough-shaped structure. Moreover, Figure 5b demonstrates the TEM and HR-TEM images.

The post-electrolysis electrode mostly consists of nanosheets with an interplanar spacing of 0.46 nm associated with CoMoO$_4$ ($\bar{2}$01), confirming there is no phase transformation in the catalyst after 20 h of HER. The same observation was substantiated by the SAED image shown in the inset of Figure 5b. Finally, Figure 5c presents the high-resolution Co 2p, Mo 3d, O1s, and Zn 2p XPS spectra after HER stability characterization. All peaks that appeared for all elements shifted to lower values of BE compared with the fresh sample (refer to Table S2 and Figure 3b). The comparison of the atomic proportion of elements on the surface before and after the durability test indicated that the amount of Co 2p (6.03% → 4.98%), Mo 3d (10.28% → 0.51%), and Zn 2p (4.3% → 0.72%) decreased meaningfully. On the contrary, the atomic proportion of O 1s after the HER stability measurement elevated to 65.01% compared to the fresh sample (51.03%). From these observations, it can be concluded that the oxidation states on the metals reduced after 20 h of continuous HER. Moreover, Mo and Zn in the vicinity of multiple oxygen defects might be more favorable metal sites for the adsorption of OH$^-$ compared to Co. The schematic illustration of the HER on the surface of the Co$_{0.5}$Zn$_{0.5}$MoO$_4$ electrode can be observed in Figure 6a.

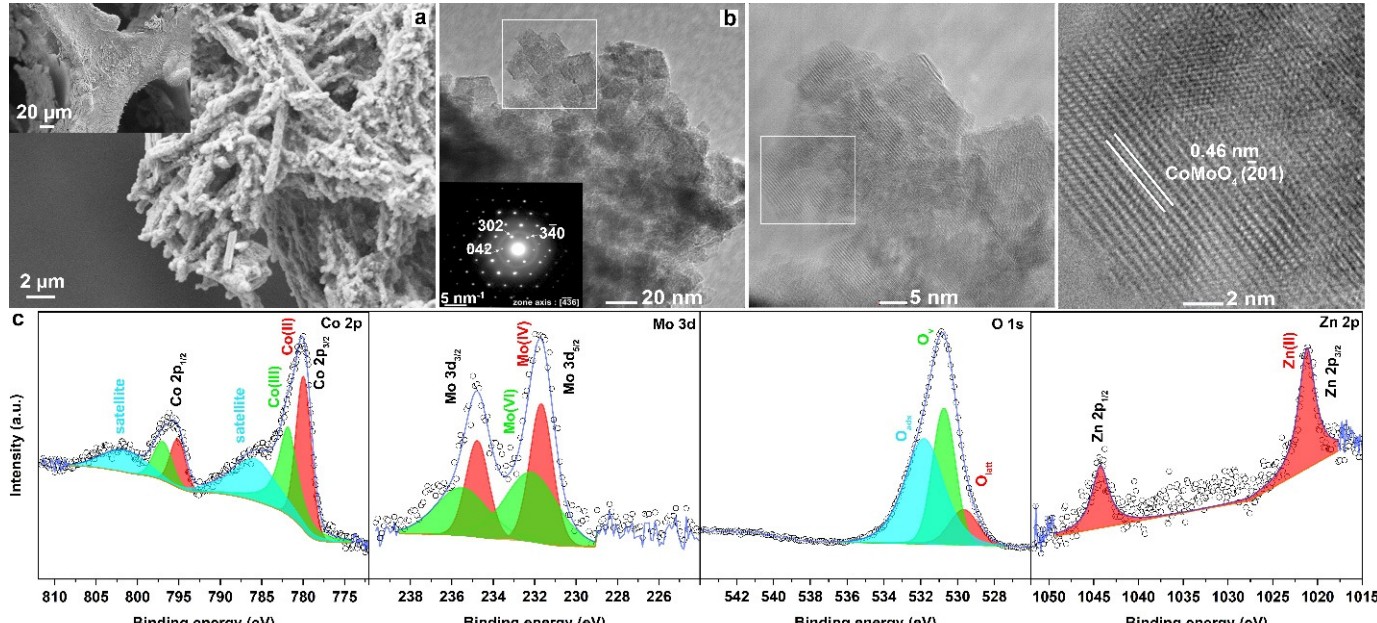

**Figure 5.** Post-electrolysis characterizations of Co$_{0.5}$Zn$_{0.5}$MoO$_4$ after long-term HER test. (**a**) SEM image (inset: low-magnification SEM image), (**b**) TEM and HR-TEM images (inset: SAED pattern), and (**c**) XPS Co 2p, Mo 3d, O 1s, and Zn 2p spectra.

The amount of H$_2$ production in a three-electrode configuration was measured at a fixed current density of 10 mA cm$^{-2}$ for 150 min at a regular interval of 30 min by a gas chromatograph, as shown in Figure 6b. The Faradaic efficiency (FE) for the HER was estimated via FE = $\frac{n_{exp}}{Q/ZF}$, in which $n_{exp}$ denotes the number of moles of H$_2$ experimentally produced at applied current density, $Q$ (C) indicates the total amount of charge, $Z$ number of transferred electrons (i.e., in HER, $Z = 2$), and $F$ is the Faraday's constant (96,485 C mol$^{-1}$). The hydrogen-production efficiency started from 94% in the first measurement and dropped to 84% in the fifth measurement (Figure 6c). The loss in Faradaic efficiency can be attributed to the partial detachment of the catalyst from the NF substrate after 150 min under exposure to 10 mA cm$^{-2}$.

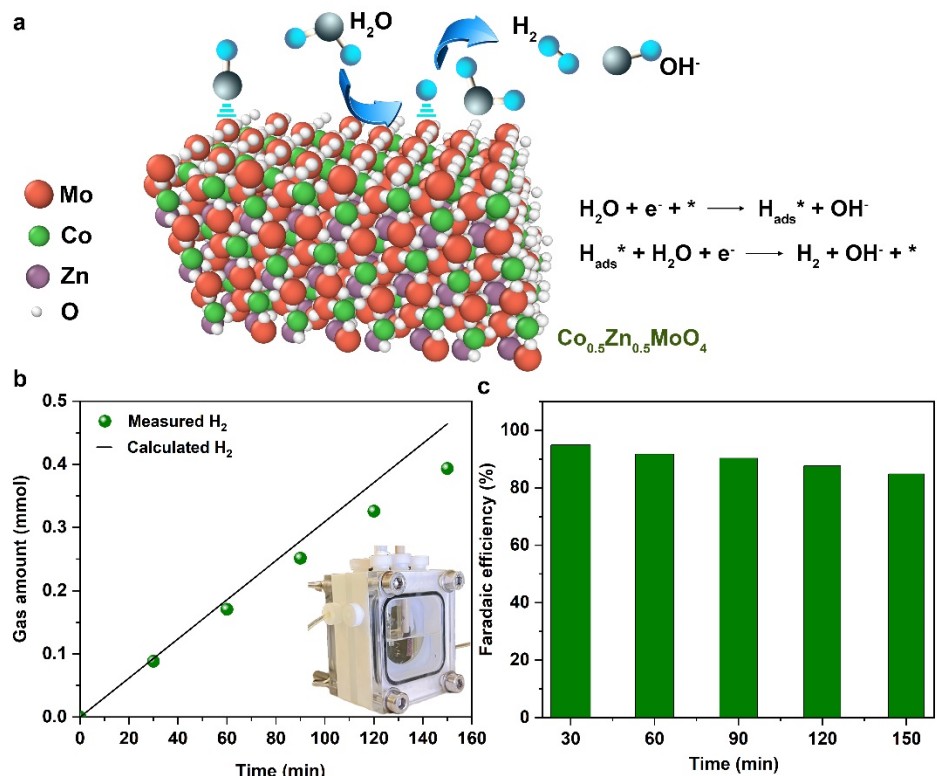

**Figure 6.** (**a**) Schematic presentation of typical Volmer–Heyrovsky pathway for alkaline HER, (**b**) Amount of gas theoretically calculated and experimentally measured for HER using $Co_{0.5}Zn_{0.5}MoO_4$ at 10 mA cm$^{-2}$ (inset: homemade electrolysis cell), and (**c**) Quantification of Faradaic efficiency derived from calculated and measured $H_2$.

## 3. Materials and Methods

### 3.1. Materials

Sodium molybdate dihydrate [$Na_2MoO_4 \cdot 2H_2O$, Sigma-Aldrich, ACS reagent, $\geq 99\%$], cobalt nitrate hexahydrate [$Co(NO_3)_2 \cdot 6H_2O$, Sigma-Aldrich, 99.999% trace metal basis], and zinc nitrate hexahydrate [$Zn(NO_3)_2 \cdot 6H_2O$, Sigma-Aldrich, 98%] were purchased and used without any prior treatment.

### 3.2. Preparation of $Co_{1-x}Zn_xMoO_4@NF$

A piece of commercial NF ($1.5 \times 3$ cm$^{-2}$) was cleaned ultrasonically with 2.0 M HCl for 10 min to remove metal oxide layers on the surface and then rinsed with ethanol, acetone, and deionized (DI) water, consecutively, each for 10 min. The growth of $Co_{1-x}Zn_xMoO_4$ ($x$ = 0.0, 0.1, 0.3, 0.5, 0.7) on NF was performed using a facile hydrothermal method (refer to Figure 1a). For the preparation of bare $CoMoO_4$ ($x$ = 0), 2 mmol $NaMoO_4 \cdot 2H_2O$ and 2 mmol $Co(NO_3)_2 \cdot 6H_2O$ were completely dissolved in 40 mL DI water through stirring to receive a red-colored transparent solution. The obtained solution was transferred into a Teflon-lined autoclave with a piece of NF and heated at 140 °C for 6 h. Once the reactor cooled down naturally, the resulting product was collected, washed several times with DI water and ethanol, and vacuum-dried at 60 °C for 24 h. Finally, the purple powders were calcined at 350 °C for 2 h to acquire β-$CoMoO_4$. For the substitution of Co with Zn, a similar procedure was applied. In this regard, varying concentrations of $Zn(NO_3)_2 \cdot 6H_2O$ were introduced to adjust the molar ratio between Co and Zn. Table S6 summarizes the defined molar ratios of initial materials for the fabrication of $Co_{1-x}Zn_xMoO_4$.

### 3.3. Preparation of Pt/C@NF Electrocatalyst

According to our previous study [9], 1 mg commercial 20% Pt/C was ultrasonically dispersed in a solution of pure ethanol (300 μL) and DI water (200 μL) for 30 min. Afterward,

10 µL Nafion solution was added and sonicated for another 30 min to gain a homogeneous ink solution. The prepared ink was dipped at once on a piece of NF to obtain a mass loading of approximately 1 mg cm$^{-2}$.

### 3.4. Materials Characterization

The crystal phase and purity of the products were analyzed by X-ray diffraction (XRD, Rigaku Mini Flex 600, Cu Kα radiation, λ = 1.5418 Å, Rigaku, Tokyo, Japan). The morphology was investigated by a field emission-scanning electron microscope (FE-SEM; Zeiss Ultra Plus, Zeiss, Jena, Germany) equipped with an energy-dispersive X-ray spectroscopy detector (EDS, Bruker Xflash 5010, 123 eV spectral resolution, Bruker Corporation, Billerica, MA, USA). The microstructure, selected area electron diffraction (SAED), and lattice fringes were further studied by high-resolution transmission electron microscopy (HR-TEM; Thermo Scientific Talos F200S 200 kV, Thermo Fisher Scientific, Waltham, MA, USA). In addition, high-angle annular dark field-scanning transmission electron microscopy (HAADF-STEM) images and the associated EDS-STEM mappings were taken on a microscope (Hitachi HF5000 200kV (S)TEM, working at HR mode, Brisbane, Australia). The surface composition and oxidation states were obtained by X-ray photoelectron spectroscopy (XPS, Thermo Scientific K-Alpha with an Al Kα monochromator source, 1486.6 eV, Thermo Fisher Scientific, Waltham, MA, USA). All XPS spectra were corrected according to the binding energy (BE) of C 1s = 284.50 eV. The functional groups were identified by Fourier transform-infrared spectroscopy (FT-IR, JASCO 6800 full vacuum and FT-IR microscope, Jasco Corporation, Tokyo, Japan). The BET-specific surface area was determined through N$_2$ adsorption–desorption isotherms (BET, Micromeritics ASAP 2010, Micromeritics Instruments Corporation, Norcross, GA, USA).

### 3.5. Electrochemical Measurements

The electrochemical properties of the prepared catalysts were investigated by conducting tests on an Autolab potentiostat/galvanostat instrument using a standard three-electrode cell. The cell consisted of a reversible hydrogen electrode (RHE; HydroFlex), Pt spring, and as-prepared catalysts on NF (0.5 cm × 1 cm) as reference, counter, and working electrodes, respectively, under 1.0 M KOH medium. To this end, the linear sweep voltammetry (LSV) curves were recorded at a scan rate of 5 mV s$^{-1}$ from 0 to −1 V (vs RHE) toward the HER. The required overpotential for the HER was calculated at a current density of 10 mA cm$^{-2}$. The cycling behavior of the catalysts was examined by recording cyclic voltammetry (CV) for 1000 cycles at a scan rate of 100 mV s$^{-1}$. The electrochemically active surface area (ECSA) was acquired through the double-layer capacitance (C$_{dl}$) with respect to the CV measured at various scan rates of 0.04, 0.06, 0.08, 0.1, 0.12, 0.14, 0.16, 0.18, and 0.2 V s$^{-1}$ across the potential window from 0.75 to 0.85 V versus RHE. The long-term stability of the best-performing catalyst was evaluated by chronopotentiometry test at 10 mA cm$^{-2}$ for 20 h. In addition, a frequency ranging from 100 kHz to 0.1 Hz with a 10 mV RMS sinusoidal modulation at 0 V was utilized to obtain the EIS curves. Finally, H$_2$ gas production for the three-electrode cell water electrolysis was detected and quantified using a gas chromatograph (7820A, GC-System, Agilent, Agilent Technologies, Santa Clara, CA, USA) equipped with a thermal conductivity detector (TCD). The experiment was carried out in a well-sealed homemade system to guarantee there was no gas leakage.

## 4. Conclusions

In brief, Zn was successfully introduced into the lattice of monoclinic CoMoO$_4$ via a one-pot hydrothermal protocol. The incorporation of Zn modulated the electronic structure of the host material, which caused the Co to shift to high oxidation states and stabilized the surface bond by the formation of oxygen vacancies. The electrocatalytic HER performance of Zn-substituted CoMoO$_4$ (Co$_{1-x}$Zn$_x$MoO$_4$ ($x$ = 0.1, 0.3, 0.5, and 0.7)) showed enhanced activity compared to parent CoMoO$_4$. Amidst the as-obtained samples of NF, the electrode with $x$ = 0.5 demonstrated the highest double-layer capacitance (C$_{dl}$ = 5.35 mF cm$^{-2}$) and

the lowest charge-transfer resistance ($R_{ct}$ = 133 Ω). In addition, $Co_{0.5}Zn_{0.5}MoO_4$ displayed the highest concentration of oxygen vacancy based on XPS results, leading to facilitated electron transfer. Due to the foregoing factors, $Co_{0.5}Zn_{0.5}MoO_4$ featured better hydrogen-evolution activity, affording 10 mA cm$^{-2}$ at an overpotential of 204 mV. Moreover, both chronopotentiometry and 1000 CV scans confirmed the excellent long-term durability of the catalyst. More importantly, the Faradaic efficiency determined the capacity of hydrogen production on the $Co_{0.5}Zn_{0.5}MoO_4$ electrode to be 10 mA cm$^{-2}$, which differed from 94% to 84%.

**Supplementary Materials:** The following supporting information can be downloaded at: https://www.mdpi.com/article/10.3390/catal13050798/s1. Figure S1. The shift of XRD (220) peaks for the prepared $Co_{1-x}Zn_xMoO_4$ samples. Figure S2. Depiction of the homogenous proliferation of catalysts on the backbone of NF substrate, (a) $CoMoO_4$, (b) $Co_{0.9}Zn_{0.1}MoO_4$, (c) $Co_{0.7}Zn_{0.3}MoO_4$, (d) $Co_{0.5}Zn_{0.5}MoO_4$, and (e) $Co_{0.3}Zn_{0.7}MoO_4$ Figure S3. (a–e) SEM/EDS and elemental mappings of $CoMoO_4$, $Co_{0.9}Zn_{0.1}MoO_4$, $Co_{0.7}Zn_{0.3}MoO_4$, and $Co_{0.5}Zn_{0.5}MoO_4$, $Co_{0.3}Zn_{0.7}MoO_4$ on NF, respectively. Table S1. EDS elemental composition of $Co_{1-x}Zn_xMoO_4$ ($x$ = 0, 0.1, 0.3, 0.5, and 0.7). Figure S4. FT-IR Spectra of prepared $Co_{1-x}Zn_xMoO_4$. Table S2. The XPS binding energy values for $CoMoO_4$, $Co_{0.5}Zn_{0.5}MoO_4$, and $Co_{0.5}Zn_{0.5}MoO_4$ after HER stability. Figure S5. (a) XPS survey, (b) Co 2p, (c) Mo 3d, (d) O 1s, and (e) Zn 2p for $Co_{1-x}Zn_xMoO_4$ ($x$ = 0, 0.1, 0.3, 0.5, and 0.7). Table S3. Comparing the HER performance of present work with previous studies. Figure S6. Typical cyclic voltammetry (CV) curves obtained at different scan rates (0.04–0.2 V s$^{-1}$) within the potential window 0.75–0.85 V vs. RHE for (a) $CoMoO_4$, (b) $Co_{0.9}Zn_{0.1}MoO_4$, (c) $Co_{0.7}Zn_{0.3}MoO_4$, (d) $Co_{0.5}Zn_{0.5}MoO_4$, and (e) $Co_{0.3}Zn_{0.7}MoO_4$. Figure S7. $N_2$ adsorption–desorption isotherms of (a) $CoMoO_4$, (b) $Co_{0.9}Zn_{0.1}MoO_4$, (c) $Co_{0.7}Zn_{0.3}MoO_4$, (d) $Co_{0.5}Zn_{0.5}MoO_4$, and (e) $Co_{0.3}Zn_{0.7}MoO_4$. Table S4. Further details derived from BET analysis of the samples. Table S5. Fit parameters for the samples derived from EIS experiments. Table S6. The defined molar ratios of initial materials for the fabrication of $Co_{1-x}Zn_xMoO_4$ ($x$ = 0, 0.1, 0.3, 0.5, 0.7).

**Author Contributions:** S.C. and E.S. synthesized the catalysts and performed the chemical and electrochemical measurements. N.S.P. helped with the interpretation of all analyses. U.U. helped with the Faradaic efficiency experiments and their interpretation. U.A. supervised the whole project from synthesis to interpretation of the results. All authors have read and agreed to the published version of the manuscript.

**Funding:** This work is supported by the Turkish Academy of Sciences—Outstanding Young Scientist Award Program.

**Data Availability Statement:** The data that support the findings of this study are available from the corresponding author upon reasonable request.

**Acknowledgments:** The authors would like to acknowledge Barış Yağcı and Amir Motallebzadeh, at Koç University Surface Science and Technology Center (KUYTAM) for their help with characterizations. Likewise, we thank Gülcan Çorapcıoğlu at Koç University Nanofabrication and Nanocharacterization Center for Scientific and Technological Advanced Research (n²STAR) for her help with HAADF-STEM analysis. Finally, we are grateful to Süleyman Tekmen from Bayburt University Central Research Laboratory (BUMER) for the HR-TEM measurements.

**Conflicts of Interest:** There are no conflicts of interest to declare.

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
