# Peer review of "Tuning Electrochemical Hydrogen-Evolution Activity of CoMoO4 through Zn Incorporation"

_catalysts, doi:10.3390/catal13050798_

Round 1
Reviewer 1 Report
The authors of this manuscript reported tuning electrochemical hydrogen evolution activity of CoMoO4 through Zn incorporation. This work might be of interesting to the readers in catalysts, as the experimental conditions used for the sample preparation and property measurements are given in detail that opens a possibility for future testing of the results by other researchers. The authors did some interesting work and the experiment data are original and abundant. My several corrections proposed for the text and questions are listed below for author consideration:
1. Why there are so many a,b,c,d in the whole manuscript. It is very confusing.
2. Many mistakes were found. Please carefully check.
3. For the stability resting in Figure 4f, where there is a sudden increase at the first two hours?
4. I did not find the reaction mechanism for the designed catalysts.
5. Is zinc ion easily soluble during the HER process?
6. For Figure S5d, THE O 1s for Co0.5Zn0.5MoO4 looks very different. Please give explanations.
7. Please compare the activity with reported materials.
8. The very important related papers should be cited, such as Composites Part B: Engineering, 2020, 198, 108214; Energy & Environmental Materials, 2022, e12441; Advanced Functional Materials, 2021, 31, 16, 2009779.
no
Author Response
Dear Editor,
We are pleased that the reviewers find our work interesting for publication in Catalysts. We carefully considered their comments, made appropriate changes to our manuscript, and highlighted the edits. We hope these changes improved our paper's conciseness to a level suitable for publication. You may find the comments together with our corresponding responses and changes, below.
With Best Regards
Asst. Prof. Umut Aydemir
Reviewer 1:
General Comment: The authors of this manuscript reported tuning electrochemical hydrogen evolution activity of CoMoO4 through Zn incorporation. This work might be of interesting to the readers in catalysts, as the experimental conditions used for the sample preparation and property measurements are given in detail that opens a possibility for future testing of the results by other researchers. The authors did some interesting work and the experiment data are original and abundant. My several corrections proposed for the text and questions are listed below for author consideration:
Response:
We would like to thank the Reviewer for taking their valuable time to evaluate the present work. The authors would like to state that the valuable comments of the Reviewer have been carefully considered, the required modifications have been done, and the relevant sections have been highlighted.
Comment 1:
Why there are so many a,b,c,d in the whole manuscript. It is very confusing.
Response:
The authors do realize the Reviewer’s concern. For this, in some figures that include TEM images or XPS spectra, a bunch of panels have been labeled by a single letter.
Comment 2:
Many mistakes were found. Please carefully check.
Response:
It should be noted that the manuscript has been meticulously developed and has undergone scrupulous revision several times.
Comment 3:
For the stability resting in Figure 4f, where there is a sudden increase at the first two hours?
Response:
It is worth noting that the potential, in the beginning, is around 220 mV vs RHE and decreases to 180 mV vs RHE after 2h. The potential drop within the first 2 h might be on account of changes in the oxidation states of transition metals and exfoliation of nanostructures on the surface, giving rise to abundant active sites on the electrode.
Comment 4:
I did not find the reaction mechanism for the designed catalysts.
Response:
According to Figure 4c, all as-prepared materials showed Tafel slope values beyond 120 mV dec–1, implying that the rate of the reaction is determined by the adsorption of hydrogen atoms on the active sites.
In addition, we believe that the hydrogen evolution on the fabricated electrodes proceeds via the following reactions:
H2O + e– + * → Hads* + OH– (Volmer step)
Hads* + H2O + e– → H2 + OH– + * (Heyrovsky step)
Comment 5:
Is zinc ion easily soluble during the HER process?
Response:
The catalysts were characterized after 20 h HER long-term durability measurement. The XPS analysis performed on the post-HER catalyst demonstrated that Zn is not soluble in the course of the reaction (refer to Figure 5c in the manuscript).
Comment 6:
For Figure S5d, THE O 1s for Co0.5Zn0.5MoO4 looks very different. Please give explanations.
Response:
Similar to CoMoO4, the core-level O 1s XPS spectrum for Co0.5Zn0.5MoO4 was analyzed, and three peaks were obtained at 530.07, 532.17, and 534.12 eV corresponding to the identical oxygen-bearing groups. As expected, the amount of oxygen vacancy mounted in Co0.5Zn0.5MoO4 relative to unsubstituted CoMoO4. The XPS spectra of all elements for each sample can be observed in Figures S5b-e.
According to the deconvoluted XPS spectra, the difference stems from the large oxygen vacancy peak associated with Co0.5Zn0.5MoO4.
Comment 7:
Please compare the activity with reported materials.
Response:
Thanks to the Reviewer’s valuable comment, we have compiled the HER performance of similar studies and summarized it in a table. Table S4 has been added to the supporting information.
Comment 8:
The very important related papers should be cited, such as Composites Part B: Engineering, 2020, 198, 108214; Energy & Environmental Materials, 2022, e12441; Advanced Functional Materials, 2021, 31, 16, 2009779.
Response:
We have cited the relevant publications in the revised version of the manuscript.
Reviewer 2 Report
This paper focuses on Zn-doped CoMoO4 electrocatalysts for hydrogen evolution reaction (HER) in alkaline media. The catalytic activity was moderate, but the catalyst was well characterized. Among the most important is the analysis of the catalyst after HER. The overall quality and novelty of this paper may be suitable for your esteemed journal. However, this paper has many errors, and the authors must revise and correct them carefully.
・It is recommended that the explanation of figures like page 5 be modified as they are difficult to understand (e.g., a to Figure 2a or (a)).
・ The superfluous line breaks should be removed.
・The reference source needs to be corrected (Error! Reference source not found).
Author Response
Dear Editor,
We are pleased that the reviewers find our work interesting for publication in Catalysts. We carefully considered their comments, made appropriate changes to our manuscript, and highlighted the edits. We hope these changes improved our paper's conciseness to a level suitable for publication. You may find the comments together with our corresponding responses and changes, below.
With Best Regards
Asst. Prof. Umut Aydemir
Reviewer 2:
General Comment: This paper focuses on Zn-doped CoMoO4 electrocatalysts for hydrogen evolution reaction (HER) in alkaline media. The catalytic activity was moderate, but the catalyst was well characterized. Among the most important is the analysis of the catalyst after HER. The overall quality and novelty of this paper may be suitable for your esteemed journal. However, this paper has many errors, and the authors must revise and correct them carefully.
Response:
We would like to thank the Reviewer for taking their valuable time to evaluate the present work. The authors would like to state that the valuable comments of the Reviewer have been carefully considered, the required modifications have been made, and the relevant sections have been highlighted.
Comment 1:
It is recommended that the explanation of figures like page 5 be modified as they are difficult to understand (e.g., a to Figure 2a or (a)).
Response:
Thanks to the Reviewer’s invaluable suggestion, we have made the appropriate changes.
Comment 2:
The superfluous line breaks should be removed.
Response:
The relevant paragraphs were combined.
Comment 3:
The reference source needs to be corrected (Error! Reference source not found).
Response:
If we may understand the comment properly, the mentioned error was not found in the current Word Doc manuscript.
Round 2
Reviewer 1 Report
Why there are so many a,b,c,d in the whole manuscript. It is very confusing.
Please carefully revise before acceptance.
no
Author Response
Dear Editor,
Once again, we are pleased that the reviewers find our work interesting for publication in Catalysts. We carefully considered their comments, made appropriate changes to our manuscript, and used track changes for the edits. We hope these changes improved our paper's conciseness to a level suitable for publication. You may find the comments, together with our corresponding responses and changes, below.
With Best Regards
Asst. Prof. Umut Aydemir
Comment 1:
Why there are so many a,b,c,d in the whole manuscript. It is very confusing.
Response:
We thank the Reviewer's valuable time for reviewing our manuscript carefully. We need to mention what we are talking about throughout the text. For this, we cite exactly what part of the figure we are referring to. This is the only way to make the reader follow the logic of the text.